# Reducing White Adipose Tissue Browning Using p38α MAPK Inhibitors Ameliorates Cancer-Associated Cachexia as Assessed by Magnetic Resonance Imaging

**DOI:** 10.3390/nu14153013

**Published:** 2022-07-22

**Authors:** Yufei Zhao, Jingyue Dai, Yang Jiang, Honghong Wu, Ying Cui, Xinxiang Li, Hui Mao, Binghui Wang, Shenghong Ju, Xin-Gui Peng

**Affiliations:** 1Jiangsu Key Laboratory of Molecular and Functional Imaging, Department of Radiology, Zhongda Hospital, Medical School, Southeast University, Nanjing 210009, China; yufei66899@163.com (Y.Z.); userdaijy@163.com (J.D.); troyjiang1008@163.com (Y.J.); hhseu520@163.com (H.W.); am_cuiy@sina.com (Y.C.); lixinxiang2020@163.com (X.L.); jsh@seu.edu.cn (S.J.); 2Department of Radiology and Imaging Sciences, Emory University School of Medicine, Atlanta, GA 30329, USA; hmao@emory.edu; 3Monash Centre of Cardiovascular Research and Education in Therapeutics, School of Public Health and Preventive Medicine, Monash University, Melbourne, VIC 3004, Australia; Bing.Wang@monash.edu; 4People’s Hospital of Lishui District, 86 Chongwen Road, Yongyang Town, Lishui District, Nanjing 211299, China

**Keywords:** cachexia, adipose tissue, pancreatic cancer, magnetic resonance imaging, p38 MAPK inhibitor

## Abstract

Background: Up to 80% of pancreatic cancer patients suffer from cachexia. White adipose tissue (WAT) browning caused by the tumorigenicity and progression aggravates the cancer-associated cachexia (CAC). Cancer-initiated changes in the protein-38 mitogen-activated protein kinases (p38 MAPK) pathway are likely involved in the development of CAC. Methods: p38 MAPK inhibitors, VCP979 or SB203580, were used in the *in vitro* and *in vivo* models of pancreatic cancer cachexia. Expression of uncoupling protein 1 (UCP1) in the p38 MARK pathway and the properties and level of white adipocytes were analyzed and correlated to browning, followed by immunohistochemistry and Western blotting validations. Changes in the volume and fat fraction of WAT in animals were monitored by magnetic resonance imaging (MRI). Results: The size of white adipocytes was increased after being treated with the p38 MAPK inhibitors, along with increase in the MRI-measured volume and fat fraction of WAT. Comparing two p38 MAPK inhibitors, the p38α subunit-specific inhibitor VCP979 had a better therapeutic effect than SB203580, which targets both p38α and β subunits. Conclusions: Blockade of p38 MAPK reduced the WAT browning that contributes to CAC. Thus, p38 MARK inhibitors can potentially be used as a therapy for treating CAC. Non-invasive MRI can also be applied to assess the progression and treatment responses of CAC.

## 1. Introduction

Cancer-associated cachexia (CAC) is a complex metabolic syndrome characterized as the loss of both adipose tissue and lean body mass that cannot be reversed by the conventional nutritional support and treatment [1,2]. Currently, the change in body weight is still one of the major criteria for diagnosis and evaluation of cachexia in cancer patients. The international consensus defines CAC based on weight loss greater than 5% over six months, or weight loss greater than 2% in individuals with a body mass index lower than 20 kg/m^2^ [3].

Among all cancers, pancreatic cancer causes cachexia at the highest rate of 80% with approximately one third of patients ultimately succumbing to cachexia-related complications [4,5]. In the early stage of pancreatic cancer, beige adipocytes in white adipose tissue (WAT) of the body fat, which has the same function as the brown adipocytes but share many features in morphology between white adipocytes and brown adipocytes, increase substantially [6]. This process, termed as “browning” of WAT is due to the overexpression of uncoupling protein 1 (UCP1) which can increase adipocyte lipolysis and metabolism, leading to increased energy expenditure, diminished treatment tolerance and eventually, shortened survival [7,8]. However, the association of cancer development and progression with the browning process of WAT and its contribution to CAC are still unclear.

p38 mitogen-activated protein kinases (p38 MAPK) are a family of four isoforms (α, β, γ, δ) of the stress response signaling molecules that are pivotal regulators of inflammation, proliferation, migration and apoptosis [9,10,11,12]. p38 MAPK, especially with the p38α isomer, are known as a central mediator of multiple pathways involving phosphorylation and the activation of transcription factor 2 that can induce the transcription of the UCP1 gene and browning of adipocytes to produce heat [13,14]. Thus, it is possible that cancer-initiated alterations in the p38 MAPK pathway may play a role in the development of CAC by promoting the browning of WAT. If so, inhibiting the p38 MAPK pathway, especially on the p38α subunit, to control and slow the process of WAT browning may potentially treat CAC.

In this study, we investigate whether the p38 MAPK pathway is involved in the browning of WAT in pancreatic cancer and its contribution to CAC by interfering with the p38 MAPK pathway using p38 MAPK inhibitors in *in vitro* and *in vivo* models of CAC. The p38 MAPK inhibitors used in the study are SB203580, which is a non-selective pyridinylimidazole-based inhibitor of both p38α and β isoforms of p38 MAPK, and VCP979, which is a tetra-substituted thiophene small molecule inhibitor specific to the p38α subunit with high binding affinity. Both inhibitors have been shown to slow the tumor progression of pancreatic cancer [15,16]. Considering multiple cell signaling factors, such as interleukin-6 and tumor necrosis factor-α, which are involved in the interplay between adipose tissue remodeling and metabolic changes in cancer patients [17], our investigation *in vitro* was performed with a cell culture assay using white adipocytes cultured with cancer-conditioned medium obtained from growing pancreatic cancer cells to provide various cancer-cell-secreted factors. Since magnetic resonance imaging (MRI), a widely used imaging tool with high spatial resolution and soft-tissue contrast, can be used to distinguish WAT and brown adipose tissue (BAT) and subsequently measure their contents in different organs [18,19], we used MRI to perform time-dependent monitoring and quantification of the WAT change as responses to the p38 MAPK inhibitor treatment of CAC in the mouse model of pancreatic cancer.

## 2. Materials and Methods 

### 2.1. Culture of Primary White Adipocytes and Pancreatic Cancer Cells

For primary white adipocytes, stroma-vascular fraction (SVF) isolated from inguinal white adipose tissues (iWAT) of mice (3–4 weeks old) was cultured and differentiated into mature adipocytes. iWAT was divided into smaller pieces and digested in phosphate-buffered saline (PBS, Hyclone, Logan, UT, USA) with 0.5 mg/mL collagenase I (Worthington Biochemical Corporation, Lakewood, NJ, USA) at 37 °C for 1.5 h. After digestion, samples were filtered sequentially through nylon filters with a pore size of 600 and 50 µm, and centrifuged at 800 rpm for 5 min. According to the typical induction cocktail method [20,21,22], the obtained SVF cell pellet was then resuspended in the complete medium, which containing 89% Dulbecco’s modified Eagle’s medium (DMEM/F12, Gibco, Grand Island, NY, USA), 10% fetal bovine serum (FBS, Gibco, Grand Island, NY, USA) and 1% penicillin and streptomycin (PS, Hyclone, Logan, UT, USA). Resuspended SVF cells were incubated in 6-well plates at a concentration of 8 × 10^4^ cells/well maintained at 37 °C in the presence of 5% CO_2_. Once the cells reached 80% confluence, they were exposed to the differentiation induction medium I, which was a complete medium consisting of 1 µM dexamethasone (DEX, Sigma-Aldrich, St. Louis, MO, USA), 5 µg/mL insulin (Sigma-Aldrich, St. Louis, MO, USA), 1 µM rosiglitazone (ROS, Sigma-Aldrich, St. Louis, MO, USA) and 0.5 µM isobutylmethylxanthine (IBMX, Sigma-Aldrich, St. Louis, MO, USA) for 72 h to boost differentiation, followed by differentiation induction medium II, i.e., complete medium with 5 µg/mL insulin and 1 µM rosiglitazone for 72 h, to promote lipid droplet formation. The procedure of culturing primary white adipocytes was then finished with incubation in a complete medium for 48 h. To confirm the adipocyte differentiation, mature adipocytes were stained by Oil Red O (Sigma-Aldrich, St. Louis, MO, USA). The details of this method can be found in the Appendix A. The supernatant of the final medium of mature primary white adipocytes was collected after the induced maturation. The collected supernatant was filtered with a 0.22 µm pore size filter and used immediately, or stored at −20 °C.

The pancreatic cancer cell line, Panc02, was purchased from Cell Bank (Shanghai Institute of Biological Science, Shanghai, China). Panc02 cells were incubated in the complete medium until they reached a confluency of 70–80% of the logarithmic growth phase. The used culture medium was replaced with the fresh complete medium and cultured for an additional 48 h. The supernatants were then collected and centrifuged at 4 °C and 2000 rpm for 10 min before being used immediately or stored at −20 °C.

### 2.2. In Vitro Cell Culture Assay

To include cancer cell signaling factors in the cell culture experiments, the *in vitro* cell culture assay was developed by culturing white adipocytes with the supernatant separated from the medium used for culturing Panc02 cells, which is the “cancer cell conditioned medium” containing various cancer-cell-secreted co-factors, referring to the supernatant of the culture medium with pancreatic cancer cells growing, as the methods reported in the literature [23,24,25,26]. Briefly, after 80% of SVF cells adhered to the surface, a differentiation induction medium, which was a mixture of 50% supernatant of the medium used for culturing Panc02 cells and 50% fresh complete medium, was used to stimulate the differentiation of the mature primary white adipocytes as a CAC model *in vitro*. The differentiation induction medium that was prepared by mixing 50% of the supernatant cultured with mature primary white adipocytes and 50% of fresh complete medium was used as a control. After 6 days of induced differentiation, the medium was replaced with complete medium and the cells continued to be cultured for 48 h. Each experiment was conducted in triplicate and repeated at least 3 times.

### 2.3. In Vitro Assays for Evaluating p38 MAPK Inhibition

The *in vitro* cell culture assay to study CAC was constructed in 6-well plates. While changing the differentiation induction medium, different concentrations (0, 1 and 5 µmol) of SB203580 (R&D systems, Minneapolis, MN, USA) or VCP979 (gift from Professor Wang Binghui of Monash University) were added as p38 MARK inhibitor treatment for 6 days. Then, the cells were cultured for 2 days with ordinary complete medium. The cells with different p38 MARK inhibitor treatment conditions were harvested for Oil Red O staining, Western blotting and immunohistochemistry (IHC) analysis to determine the effect of the treatment based on the amount of primary white adipocytes and the level of UCP1 expression. Each experiment was repeated at least 3 times. More details on the experimental validation are described in the Appendix A.

### 2.4. p38 MAPK Inhibitor Treatment for Mice with Pancreatic Cancer

All animal studies have been approved by the appropriate ethics committee of the Medical School of Southeast University (SYXK2016-0014). The use of experimental animals followed the *3*R (reduction, replacement, refinement) principles. Because previous studies reported that female animals exhibit estrogen-driven protective responses to the metabolic perturbations and CAC development [27,28], we only used male C57BL/6J mice in this preliminary study to exclude this potential interfering factor. Animals were housed at a room temperature of 22 ± 0.5 °C on a 12 h light and 12 h dark cycle with pair-feeding and free access to water.

The orthotopic pancreatic tumor model was prepared based on the procedure reported in the literature [29,30,31,32] and as described in the Appendix A. Starting on day 1 after tumor implantation, mice (male, age 7–8 weeks old)-bearing orthotopic pancreatic tumors were randomly divided into three groups (*n* = 6/group) for studying with the p38 MARK inhibitor treatment. Animals were given intraperitoneal injections of SB203580 (PC-SB) or VCP979 (PC-VCP) at a dose of 5 mg/kg per day for 16 days, whereas those in the control group were injected with an equal volume of vehicle containing 90% saline + 10% Dimethyl sulfoxide (PC-NS). Body weight of the animals were recorded daily. The survival days of the mice in each group were documented.

### 2.5. MRI Scan and Image Analysis

Tumor-bearing mice and healthy control mice were scanned using a preclinical 7T MRI scanner (Bruker PharmaScan MRI, Ettlingen, Baden-Württemberg, Germany) with a transmit-receiver quadrature volume coil on the 16th day after starting the p38 MAPK inhibitor treatment, considering the need to observe the measurable therapeutic effect. Mice were anaesthetized by the inhalation of 0.5–1% isofluorane (Keyuan, Jinan, Shandong, China) mixed in oxygen during MRI scans. The scanning parameters of MRI and image analysis were performed according to the previously reported method [18,33]. T1 weighted images (T_1_WI) were acquired first using a multi-slice multi-echo sequence in the axial direction with scanning parameters as: time of repetition (TR) of 500 ms, time of echo (TE) of 15 ms, flip angle of 180°, number of excitations of 2, field of view (FOV) of 4 × 4 cm^2^, matrix of 256 × 256, number of slices of 16, and slice thickness of 1 mm with a 0.6 mm gap.

Two series of chemical shift-selective imaging (CSSI), one tuned at the chemical shift of fat protons (3.5 ppm) and the other one at water protons (4.5 ppm), were performed using a rapid acquisition with a relaxation enhancement sequence. In this case, fat selective images were obtained by placing the center of the excitation bandwidth at the resonance frequency of fat (−1000 Hz) downfield from the water resonance frequency (0 Hz) which is the center frequency at 7T. To achieve the chemical-specific detection of selected fat or water signals, the gradient field was controlled during the 180° pulse to only refocus on the selected fat or water protons. CSSI data acquisition parameters included: TR of 1000.0 ms, TE of 9.9 ms, flip angle of 180°, 4 excitations, FOV of 4 × 4 cm^2^, matrix of 256 × 256 and slice thickness of 1 mm.

The volumes of subcutaneous WAT (sWAT), visceral WAT (vWAT), total WAT (tWAT), tumor and ascites were measured using T_1_WI by ImageJ software (National Institutes of Health, Bethesda, MD, USA) in the manually traced regions of interest (ROI). For analyzing CSSI data, ROIs was placed at the same site of the WAT in both fat and water images, respectively. ROIs containing 15–20 pixels (about 0.25 mm^2^) placed on sWAT in the inguinal region, epididymal WAT (eWAT) and perirenal adipose tissue were chosen as the representatives of vWAT. The signal intensities (SI) in the ROIs were measured using a program (ParaVision 5.0; Bruker PharmaScan, Ettlingen, Baden-Württemberg, Germany) from both fat selective images (SI_fat_) and water selective images (SI_water_). The fat fractions of different WAT were calculated following the equation below: FF_CSSI_ = 100 × SI_fat_/(SI_fat_ + SI_water_ × R)(1)
where R is the ratio of the fat-to-water proton densities in their pure form, which has a value of 0.9 as used in the early publication [34].

### 2.6. Histopathologic Analysis and Western Blotting

After 16 days of treatment and their last MRI scans, mice were sacrificed on day 17 to collect adipose tissue and liver samples for histological analysis (*n* = 6/group).

Tissue samples were perfused with PBS and 4% paraformaldehyde (Sigma-Aldrich, St. Louis, MO, USA). The sections of sWAT, vWAT and liver were carefully stripped out, then fixed in formalin for 96 h and embedded in the paraffin wax. Fixed tissue sections were cut into 4 μm slices and then mounted on a glass slide before being stained using hematoxylin-eosin staining (H&E, Sigma-Aldrich, St. Louis, MO, USA).

Portions of sWAT and vWAT samples were separated for IHC staining from mice in different groups. sWAT and vWAT sections were treated with the citrate buffer (pH 6.0) for 45 min at 96 °C for antigen retrieval and then blocked using the peroxidase and protein block solution (Novolink Polymer Detection Systems Kit, Leica Biosystem, Newcastle, UK). Subsequently, the sections were incubated overnight at 4 °C with UCP1 primary antibody (1:500 dilution, Abcam, Cambridge, MA, USA) and Novolink Polymer (Leica Biosystems, Wetzlar, Hessian, Germany).

Western blotting was performed to detect the protein expression of sWAT and eWAT collected from mice with and without the p38 MAPK inhibitor treatment. Proteins were extracted from the tissues with the protein extraction buffer (RIPA, Keygen Biotech, Nanjing, Jiangsu, China) in the presence of phosphatase inhibitors (10% PMSF, Keygen Biotech, Nanjing, Jiangsu, China). After protein concentrations were determined using the bicinchoninic acid assay protein quantification kit (Keygen Biotech, Nanjing, Jiangsu, China), 30 μg of proteins from each tissue sample was separated by the sodium dodecyl sulphate polyacrylamide gel electrophoresis and blotted, and then transferred to the polyvinyl difluoride membranes. The membranes were then incubated with antibodies including UCP1, Anti-β-actin, p38 and p-p38 (1:1000 dilution, Abcam, Cambridge, MA, USA) at 4 °C overnight, followed by incubation with a horseradish peroxidase-conjugated secondary antibody (1:5000 dilution, Abcam, Cambridge, MA, USA).

Quantitative analysis of the H&E, IHC staining and Western blotting results was performed using ImageJ analysis software (National Institutes of Health, Bethesda, MD, USA).

### 2.7. Statistical Analysis

All statistical analyses were performed using SPSS software (SPSS for Windows, version 11.0, 2001; SPSS, Chicago, IL, USA). Numerical data from multiple measurements were presented as the mean ± standard deviation (SD). One-way analysis of variance was used for analyzing the data from the *in vitro* experiments for evaluating the protein expression obtained from Western blotting in different conditions. Independent-sample t-test was performed to analyze and compare the data from the *in vivo* experiments comparing SB203580, VCP979 treatment and control groups, including the body weight, food intake, volume and FF of WAT measured by MRI. The results were correlated to those from histopathologic analysis and Western blotting.

For statistical comparisons, a *p* value of less than 0.05 was considered to indicate a statistical difference.

## 3. Results

### 3.1. Pancreatic Cancer Promoting Lipid Metabolism That Leads to CAC

Primary white adipocytes are capable of producing fatty lipids under the condition of the adipogenic cocktail, as Oil Red O-stained lipid droplets with different sizes were found in those cells after 6 days of induction and 2 days of incubation in the complete medium (Figure 1A). After adding the culture supernatant from growth media containing Panc02 cells growing in the logarithmic growth phase to the primary white adipocytes, we observed a sharp decrease in lipid droplets in the Oil Red O stain (*p* < 0.01) (Figure 1B).

When comparing the body weight of tumor-bearing mice with healthy control mice (*n* = 6/group), we found no significant difference in the overall body weight with tumors included (26.14 ± 2.10 g) and healthy control animals (26.48 ± 1.76 g) (*p* = 0.79) (Appendix A). After the tumors and ascites were removed, the mean adjusted body weight of tumor-bearing mice was 23.62 ± 0.82 g, which is significantly lower than that of the healthy mice (*p* < 0.01) (Figure 1C). In comparison to the baseline body weight before the tumor implantation, mice with pancreatic tumors exhibited a significant decrease in body weight (−1.41 ± 1.03 g) (*p* < 0.001), or > 5% of their body weight (5.97 ± 1.58%), likely due to cachexia (Figure 1D), while the weight of the healthy animals in the control group gained an average of 3.10 ± 0.99 g. When measuring the fat content at 21 days after tumor implantation using T_1_WI, we found that the volumes of sWAT, vWAT and tWAT deposits in tumor-bearing mice were 0.21 ± 0.02, 0.16 ± 0.03 and 0.50 ± 0.03 cm^3^, respectively, all of which were significantly lower than those of the control group, i.e., 1.50 ± 0.21 (*p* < 0.001), 1.11 ± 0.20 (*p* < 0.001) and 3.52 ± 0.24 cm^3^ (*p* < 0.001) (Figure 1E).

H&E staining revealed that the size of the white adipocytes of sWAT and the size of lipid vesicles in tumor-bearing mice were smaller than those of the controls, and gradually decreased from week 1 to 3 as pancreatic cancer advanced further. Furthermore, the degradation of fat with the excessive macro- and micro-vesicular fat deposits in the liver tissue was also observed, indicating potential ectopic deposition of triglycerides and the degree gradually increased as the CAC was worsened over the three weeks of experiments (Figure 1F).

### 3.2. Increased WAT Browning in CAC

To examine whether the browning of white adipocytes regulated by UCP1 and p38 plays a role in the development of CAC, we first performed the Western blotting and IHC analyses to determine the expression level of UCP1 and phosphorylated p38 (p-p38) and their changes in the *in vitro* cell culture assay with primary white adipocytes growing in the cancer cell-conditioned medium (Appendix A). The results from quantifying the expression data from Western blotting (Appendix A) showed significantly increased levels of UCP1 and p-p38/p38 of white adipocytes in the cancer cell-conditioned medium compared to the controls *in vitro* (*p* < 0.01, *p* < 0.001) (Figure 2A).

Since the CSSI-based MRI method is capable of differentiating different fat contents, sWAT, eWAT and vWAT, in the selected tissue compartments of animals, can be measured non-invasively, as shown in Figure 2B. The results revealed that sWAT, eWAT and vWAT in mice with pancreatic cancer are 69.45 ± 3.25, 67.06 ± 3.21 and 74.04 ± 0.62%, respectively, compared to 83.66 ± 3.38, 82.49 ± 0.92 and 84.20 ± 1.30% in those of healthy controls (*n* = 6/group) (Figure 2C). Consistent with the observations in cell culture assays, IHC staining of adipose tissue samples collected from tumor-bearing mice at 1, 2 and 3 weeks after tumor inoculation confirmed the browning of white adipocytes taking place in mice with pancreatic cancer, while the UCP1 expression level increased gradually with the progression of pancreatic cancer (Figure 2D–E). To further validate the MRI findings, we performed the Western blotting analysis on the collected tissue samples. We observed that the levels of UCP1 expression and p38 phosphorylation in iWAT (*p* < 0.01, *p* < 0.001) and vWAT (*p* < 0.001, *p* < 0.001) of mice with pancreatic cancer were significantly increased (Appendix A, Figure 2F), compared to those obtained from the control animals.

### 3.3. Effect of P38 MAPK Inhibitors on the Browning of White Adipocytes and CAC 

Since UCP1 and p38 phosphorylation are key components of the p38 MAPK pathway and were found to be elevated in the WAT of mice with pancreatic cancer in this study, we further tested whether p38 MAPK inhibitors may block the browning of white adipocytes and slow CAC. We first cultured primary white adipocytes with the pancreatic cancer cell-conditioned medium with or without adding one of the two p38 MAPK inhibitors, i.e., VCP979 or SB203580, at different dosages, followed by measuring the size of lipid droplets stained by the Oil Red O (Figure 3A). We observed that the size of the lipid droplets in the white adipocytes treated with p38 inhibitors were bigger than those in the samples with no p38 inhibitor treatment (Appendix A). Furthermore, Oil-Red-stained lipid droplets were more intense in the samples treated with a high dosage of VCP979 or SB203580 (5 µmol) compared to those treated at a low dosage (1 µmol). IHC staining of UCP1 proteins in white adipocytes treated with different p38 MAPK inhibitors at different dosages also showed similar patterns of responses with UCP1 proteins reduced after p38 MAPK inhibitor treatment (Figure 3B). Analyzing data from the Western blotting of the samples under different treatment conditions (Figure 3C) revealed that the expression of UCP1 and the level of p38 phosphorylation were all downregulated by the p38 MAPK inhibitor treatment in a dose-dependent manner (Figure 3D). Moreover, VCP979, which is the p38α subunit-specific inhibitor, appears to have a stronger effect on downregulating UCP1 than SB203580, which targets both p38α and β subunits.

We then performed the *in vivo* assessment of the treatment responses of p38 MAPK inhibitors in the mouse model of pancreatic cancer following the scheme shown in Figure 4A. We found that the average body weights of the VCP979-treated group (PC-VCP) and SB203580 group (PC-SB) were 24.35 ± 1.50 and 23.16 ± 1.61 g after 16 days of continuous p38 MAPK inhibitor treatment (Appendix A), compared to 23.81 ± 0.93 g of those in the group with no treatment (PC-NS). The body weight of the PC-VCP group showed slightly higher results. However, there is no statistical difference between each of the three groups (*p* = 0.67).

Since MRI methods are capable of non-invasively identifying, differentiating and measuring volumes of the adipose tissue, tumors and ascites using T_1_WI (Figure 4B) and CSSI (Figure 4D) techniques, the development of CAC and responses to the p38 MAPK inhibitor treatment can be monitored in the same mice at different time points. Table 1 summarizes the results of the MRI-measured volumes of adipose tissue, tumors and ascites obtained from the different groups on day 8 and 16. When we compared the volumes obtained from the different groups, it was found that sWAT and vWAT were similar among the three groups on the eighth day of treatment except tWAT of p38 MAPK inhibitor-treated PC-VCP and PC-SB groups, which were statistically significantly higher than that of the control group. In addition, we found no statistically significant difference in the tumor and ascites volume among the three groups on the eighth day of treatment (Figure 4C). After 16 days of treatment, sWAT, vWAT and tWAT in the p38 MAPK inhibitor-treated groups were significantly higher than those of animals in the PC-NS group. Ascites volumes in the PC-NS group were significantly higher than those of the VCP979-treated group (*p* < 0.05) and SB203580-treated group (*p* < 0.05). In addition, the tumor volume in the mice without treatment was slightly higher. Consistent with the previously reported results [15], there is no statistical difference between the PC-VCP and PC-SB groups regarding tumor volume measurement. Notably, the WAT volume of VCP979 was larger than that of the SB203580 treatment group (Table 1, Figure 4C). Taken together, although SB203580 and VCP979 exhibited similar therapeutic effects, the VCP979 showed a better performance in preventing the pancreatic cancer-induced fat mass decrease in all adipose tissues.

Table 2 summarizes fat fractions of sWAT, eWAT and vWAT calculated from the fat content measurement using CSSI data. After 8 consecutive days of treatment with p38 MAPK inhibitors, tumor-bearing mice treated with VCP979 (PC-VCP) exhibited higher averaged fractions of eWAT and vWAT than those treated with SB203580 (PC-SB) or without treatment (PC-NS). There is no statistically significant difference in the fraction of sWAT between the three groups. However, while all mice bearing pancreatic tumors exhibited a loss of fat fractions from day 8 to day 16 (Appendix A), the fractions of sWAT, eWAT, vWAT of the PC-VCP group were significantly higher than those of the PC-NS and PC-SB groups after 16 days of treatment (Figure 4E).

Based on the overall body weight measurement, animals with pancreatic cancer in each group suffered different degrees of weight loss (Figure 5A) compared to the averaged body weight of approximately 30 g for healthy mice. Table 3 and Figure 5B summarize the net body weight, changes in the net body weight and percentage change in net body weight of tumor-bearing mice after 16 days of treatment with saline, VCP979 or SB203580. Animals in the PC-VCP group showed statistically significantly less weight loss than those without treatment (PC-NS). When the tumors and ascites were removed, the mean adjusted body weight of mice treated with saline (PC-NS) was 1.61 ± 0.54 g less than the overall body weight, or 6.71 ± 1.18% weight loss, while animals in the p38 MAPK inhibitor-treated PC-VCP and PC-SB groups did not show significant weight loss, indicating that the p38 MAPK inhibitor treatment may help tumor-bearing mice resist the weight loss caused by CAC. H&E staining of the adipose tissue collected from the animals at the end of the experiments (Figure 5C) revealed that the size of the lipid vesicles in the sWAT and vWAT of the animals that were treated with VCP979 or SB203580 appears to be significantly larger than those in the un-treated group after 16 days of p38 MAPK inhibitor treatment. Furthermore, animals treated with VCP979 retained more sWAT and vWAT than those treated with SB203580 (Figure 5D). The results from the Western blotting show that the expression level of UCP1 protein in sWAT and vWAT was significantly reduced in the p38 MAPK inhibitor-treated groups compared to animals without treatment (*p* < 0.05, *p* < 0.05; *p* < 0.01, *p* < 0.01, respectively). We also found that the p-p38 levels showed the same trend as the expression level of UCP1 protein (*p* < 0.001, *p* < 0.001; *p* < 0.05, *p* < 0.05, respectively). Notably, VCP979 exhibited a better inhibitory effect than SB203580 based on the expression levels of UCP-1 and p-p38 (Figure 5E). When we examined the outcomes of the p38 MAPK inhibitor treatment, it was found that the median survival rate of the PC-VCP group and PC-SB group was 30.5 and 23 days, respectively, significantly longer than the 19-day median survival rate observed for the PC-NS group (*p* < 0.001; *p* < 0.05) (Figure 5F). At this timepoint, the averaged tumor sizes of SB203580- and VCP979-treated animals were similar, although VCP979 showed a better performance in inhibiting the browning of pancreatic cancer mice. Overall, the treatment with p38 MAPK inhibitors can effectively prolong the survival of animals with pancreatic cancer, with VCP979 exhibiting an improved efficacy over SB203580 (*p* < 0.01) (Figure 5F).

## 4. Discussion

Although there is an increased interest in investigating and understanding cachexia in the field of oncology, most current research and approaches have been dependent on observing the loss of muscle and fat [35]. The process and mechanism of the WAT atrophy as a result of WAT browning and its involvement in CAC are not well understood [1,5]. This work explores the mechanism of CAC for the potential treatment of CAC that can be coupled or combined with the traditional chemo- or radiation treatments to improve the outcome and survival of cancer patients [1,36]. This is particularly important in patients suffering from pancreatic cancer, as CAC is strongly associated with the failure of current treatments.

While it is known that the beige adipocyte is closely related to the ability to resist body weight gain and fat accumulation [37], weight loss due to the browning of adipocytes in cancer patients has been suggested as a contributing factor to CAC [38,39,40]. Some studies demonstrated that vWAT is largely resistant to browning, while others showed that there are increased expressions of genes involved in browning in animal models of cancer and cancer patients, [41,42]. When tumors are developing, cytokines, hormones, neuropeptides/neurotransmitters and tumor-derived factors can play a role in alliterating the metabolism of the adipose tissue throughout the body, such as inducing mitogen-activated protein kinases and the mitogen-activated protein kinase cascade reaction, thereby activating the p38 MAPK pathway [43,44]. Luan Y et al. reported that mice with CAC showed significant activation of the p38 MAPK pathway and increased thermogenesis in the adipose tissue with high UCP1 expression [45]. It was also reported that lipolysis induced by pancreatic cancer exosomes could be attenuated in the presence of the p38 MAPK inhibitor (SB203580) in 3T3-L1 and human adipocytes [46], suggesting that p38 MAPK inhibitors help slow the lipolysis of fat. P38α, which is the best-characterized member of the p38 family, is known to phosphorylate several key proteins involved in the lipid metabolism [47]. Studies by Matesanz N et al. [48] and Zhang S et al. [49] showed that the p38α acts by the PKA/CREB/UCP1 signaling pathway, in which CREB could regulate the expression of PKA C at the transcriptional level. In this mode, PKA and CREB form a positive feedback loop that can activate a thermogenic program responsible for the white-to-beige phenotypic switching with p38α functioning as a brake of the PKA signaling pathway in WAT. Therefore, the p38α subunit may play an important role in promoting the WAT browning in CAC. At the same time, the p38α subunit regulates the proliferation, differentiation and survival of many types of tumors, including pancreatic cancer [50]. Unlike VCP979, SB203580 competitively but non-selectively binds to the ATP-binding pockets of p38α and β isoforms of p38 MAPK. Given the similarity of the ATP-binding site in different kinases, SB203580 can also bind with non-p38 protein kinases to lose the specificity to the p38 protein kinases. Therefore, it is likely that high specificity and binding affinity of VCP979 to the active phosphorylated form of p38α leads to a less off-target effect [51]. A previous study by our group has shown that both p38 inhibitors, SB203580 and VCP979, have similar antitumor efficacy on mice with pancreatic cancer [15]. However, further investigation from this work suggests that the p38α-specific inhibitor VCP979 has a more prominent effect on inhibiting adipocyte differentiation and ameliorating CAC. Importantly, our results are consistent with those reported by Felipe Henriques et al. [52] and Han et al. [53], all supporting that the suppression of adipose tissue browning induced by CAC can increase survival and p38α could serve as a novel druggable target to combat metabolic diseases such as cachexia.

We also showed that the MRI-measured fat fraction in the white adipose tissue gradually decreased as conditions of pancreatic cancer get worse. In addition to routine MRI exams for assessing tumor prognosis and treatment responses in patients, this non-invasive imaging tool can be readily used to quantitatively evaluate spatial distributions of fat and its specific sub-types based on the difference in the hydration of different forms of adipocytes [54,55]. Compared with traditional invasive molecular biology and histopathological analysis, MRI-based adipose tissue quantification is a clinically feasible approach to longitudinally evaluate and monitor the progress of browning of WAT in patients with CAC and responses to new treatments, such as the p38 MAPK inhibitor treatment demonstrated in this study.

The current study has several limitations. Firstly, cachexia is a multifactorial syndrome caused by the pro-inflammatory signals and tumor-derived catabolic factors; thus, we did not verify the effect of VCP979 on various metabolic functions associated with browning. Systematic investigation regarding other contributing factors and molecular pathways may help create a generalizable strategy for the treatment of CAC caused by other cancers. Secondly, to exclude the possible effect of estrogen on the fat metabolism that may interfere with our observation of the effect from CAC, we only used male mice to understand the effects from other factors contributing to cachexia and related metabolic alterations in this study. Therefore, female animals and their hormonal effect will be included in future investigations. Thirdly, we only used a dose of 5 mg/kg inhibitors in the 16-day treatment, which was reported in a previous study [56]. Although under this treatment regime we observed that p38 MAPK inhibitors effectively and significantly slowed down disease progression, the timing and dose may not have been optimal for treating CAC. In addition, the present work only demonstrated changes at the morphological and molecular levels regarding the browning of WAT with CAC mice; therefore, more evidence is needed to understand whether WAT browning is involved in BAT thermogenesis-induced cachexia [57,58], such as physiological changes involved in the p38 MAPK pathway and BAT itself in animal models of CAC or patients. Our future follow-up studies will aim to provide a better understanding of the effect of p38 MAPK inhibitors on WAT browning associated with CAC and tumor metabolic processes.

## 5. Conclusions

In conclusion, the results from this study showed that the activation of the p38 MAPK pathway induces the browning of white adipose tissue and subsequently promotes CAC. The p38 MAPK inhibitor treatment is a potential therapy for CAC and improved overall survival, with the small molecule p38α specific-inhibitor, VCP979, exhibiting better efficacy. Furthermore, non-invasive MRI, a routine cancer imaging modality, can be applied to the diagnosis and quantitative assessment of CAC as well as the monitoring of the response to CAC treatment through differentiation and measurements of specific types of fat and their fractions in the selected organs and tissues.

## Figures and Tables

**Figure 1 nutrients-14-03013-f001:**
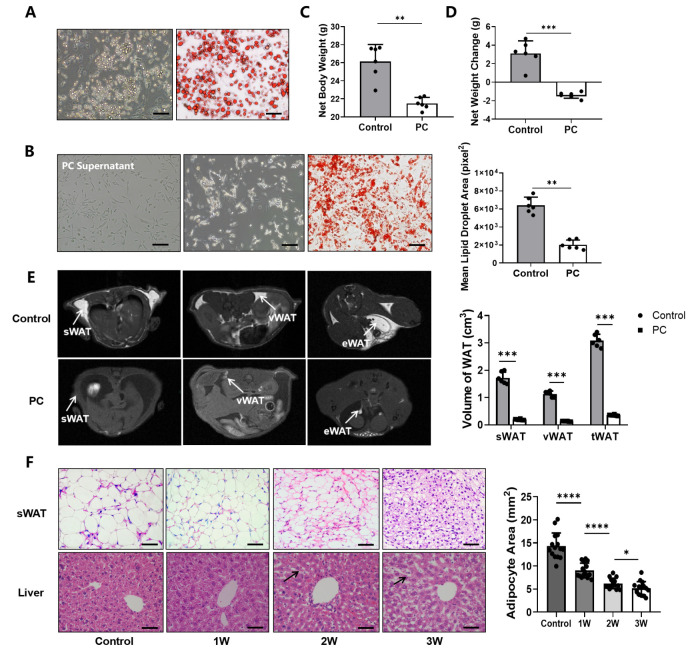
Pancreatic cancer induces abnormal fat metabolism and cachexia in the cell culture and animal model. (**A**) The mature primary white adipocytes after induction and identified by Oil Red O staining. (**B**) Panc02 cells in the left photo, primary white adipocytes in the middle photo and Oil Red O-stained primary white adipocytes in the right photo, followed by quantitative analysis of lipid droplets. (**C**) The weight of the control group and the net weight of the PC group after tumors and ascites were removed. (**D**) Changes in net body weight compared to baseline body weight. (**E**) T_1_WI MR images of healthy control mice and mice with pancreatic cancer and MRI-measured volumes of sWAT, vWAT and tWAT. (**F**) H&E staining of sWAT and liver in healthy control mice and pancreatic cancer mice as the disease progresses. (* *p* < 0.05, ** *p* < 0.01, *** *p* < 0.001, **** *p* < 0.0001). H&E = hematoxylin-eosin, iWAT = inguinal white adipose tissue, MRI = magnetic resonance imaging, PC = pancreatic cancer, sWAT = subcutaneous white adipose tissue, T_1_WI = T1 weighted image, tWAT = total white adipose tissue, vWAT = visceral white adipose tissue, WAT = white adipose tissue.

**Figure 2 nutrients-14-03013-f002:**
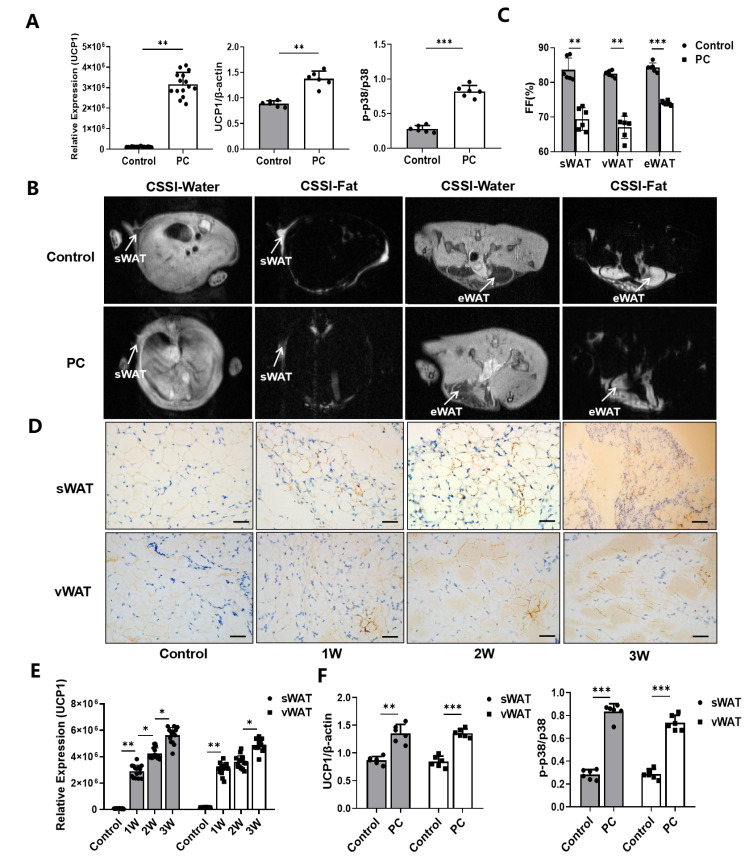
Browning of WAT and associated CAC. (**A**) The expression levels of UCP1, p-p38 and p38 protein of white adipocytes cultured with or without the Panc02 cancer cell-conditioned medium. (**B**) CSSI MRI of healthy control mice and mice with pancreatic cancer. (**C**) Measurement of the FF% of sWAT, vWAT, eWAT and tWAT. (**D**) Immunohistochemical staining of UCP1 protein in sWAT and vWAT of control mice and PC mice as the disease progresses. (**E**) The levels of expression of UCP1 by immunohistochemical staining Western blotting. (**F**) The levels of expression of UCP1, p-p38 and p38 protein of sWAT and vWAT in control mice and PC mice by Western blotting. (* *p* < 0.05, ** *p* < 0.01, *** *p* < 0.001). CAC = cancer-associated cachexia, CSSI = chemical shift-selective imaging, FF = fat fraction, MRI = magnetic resonance imaging, p- = phosphorylated, PC = pancreatic cancer, sWAT = subcutaneous white adipose tissue, tWAT = total white adipose tissue, UCP1 = uncoupling protein 1, vWAT = visceral white adipose tissue, WAT = white adipose tissue.

**Figure 3 nutrients-14-03013-f003:**
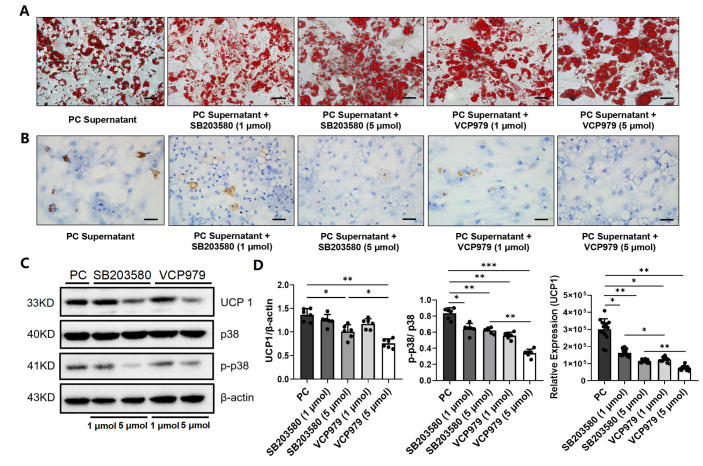
The effect of p38 MAPK inhibitors on the expression of UPC1 and levels of p-38, p-p38 and lipid droplets. (**A**) Oil Red O staining of primary white adipocytes under five different treatment conditions. (**B**) The IHC staining of UCP1 protein in primary white adipocytes under five different intervention conditions. (**C**) The Western blotting of the expression level of UCP1. (**D**) The expression level of UCP1, p38 and p-p38 in primary white adipocytes under five different intervention conditions. (* *p* < 0.05, ** *p* < 0.01, *** *p* < 0.001). p- = phosphorylated, PC = pancreatic cancer, UCP1 = uncoupling protein 1.

**Figure 4 nutrients-14-03013-f004:**
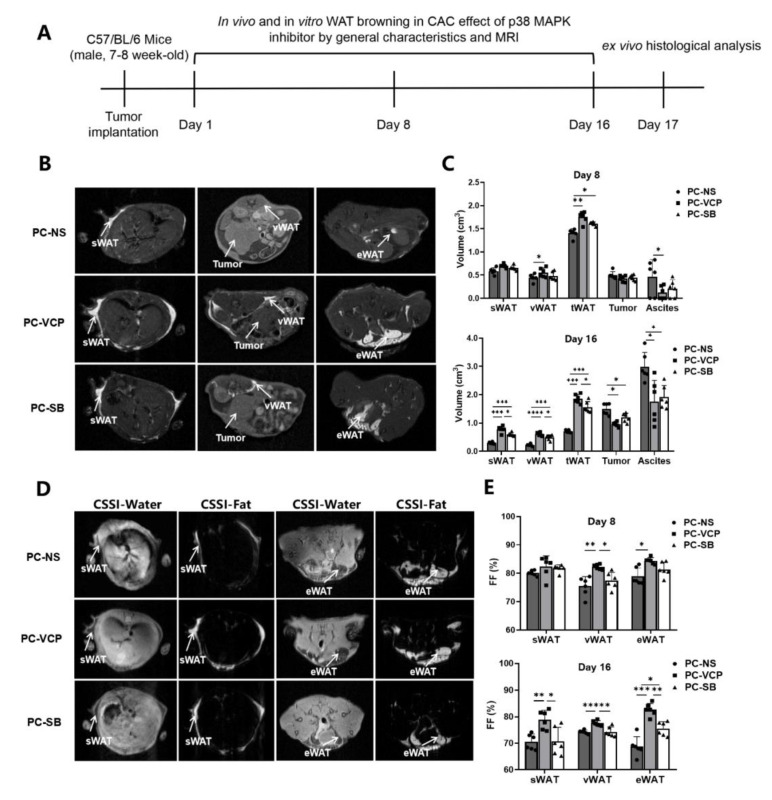
Monitoring the therapeutic effect of p38 MAPK inhibitors in mice with pancreatic cancer. (**A**) Scheme of experiments for tracking and the evaluation of treatment. (**B**) Examples of T_1_WI of selected mice in different groups. (**C**) MRI-measured volumes of sWAT, vWAT, tWAT, tumors and ascites on the 8th and 16th day after modeling. (**D**) Examples of CSSI of selected animals in different groups. (**E**) CSSI measured FF% of sWAT, vWAT and eWAT on the 8th and 16th day after starting the treatment. (* *p* < 0.05, ** *p* < 0.01, *** *p* < 0.001, **** *p* < 0.0001). CAC = cancer-associated cachexia, CSSI = chemical shift-selective imaging, eWAT = epididymis white adipose tissue, FF = fat fraction, MRI = magnetic resonance imaging, p38 MAPK = protein-38 mitogen-activated protein kinase, PC = pancreatic cancer, sWAT = subcutaneous white adipose tissue, T_1_WI = T1 weighted image, tWAT = total white adipose tissue, vWAT = visceral white adipose tissue, WAT = white adipose tissue.

**Figure 5 nutrients-14-03013-f005:**
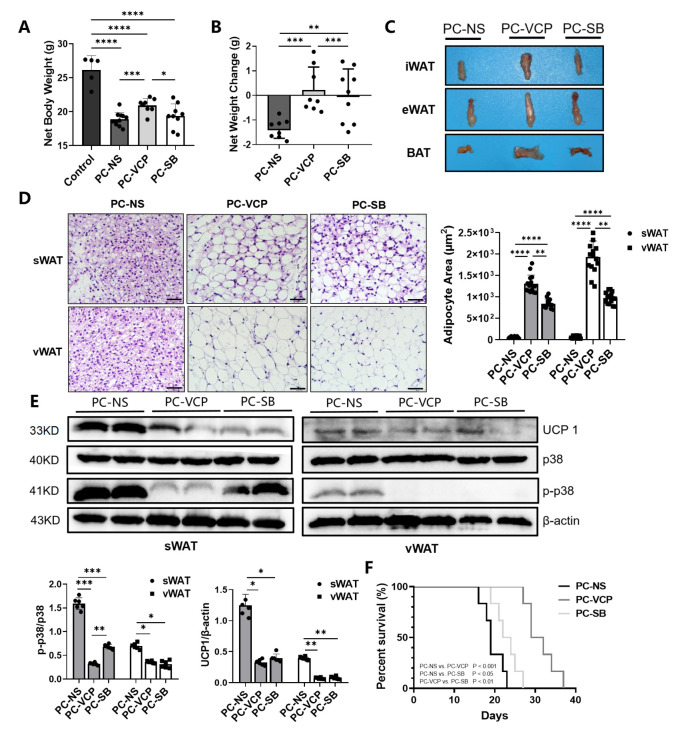
Histological validations of the effect of the p38 MAPK targeted treatment of CAC. (**A**) Comparison of the net weights of the mice in the three different groups. (**B**) Changes in the net weight of the mice in three different groups. (**C**) Representative iWAT, eWAT and BAT collected from the selected mice of each group. (**D**) H&E staining of sWAT and vWAT and the results from the measurement of adipocytes in the adipose tissue samples collected from the animals in different groups. (**E**) The Western blotting for UCP1 in tissue samples collected from the animals in different groups and their expression levels of UCP1, p38, p-p38 protein of sWAT and eWAT. (**F**) Survival curves of the three groups. (* *p* < 0.05, ** *p* < 0.01, *** *p* < 0.001, **** *p* < 0.0001). BAT = brown adipose tissue, eWAT = epididymis white adipose tissue, H&E = hematoxylin-eosin, iWAT = inguinal white adipose tissue, p- = phosphorylated, p38 MAPK = protein-38 mitogen-activated protein kinase, PC = pancreatic cancer, sWAT = subcutaneous white adipose tissue, UCP1 = uncoupling protein 1, vWAT = visceral white adipose tissue, WAT = white adipose tissue.

**Table 1 nutrients-14-03013-t001:** MRI-measured volumes of adipose tissue, tumors and ascites.

	Day 8			Day 16		
	PC-NS	PC-VCP	PC-SB	PC-NS	PC-VCP	PC-SB
sWAT (cm^3^)	0.59 ± 0.07	0.70 ± 0.04	0.66 ± 0.05	0.30 ± 0.03	0.78 ± 0.11 ***	0.60 ± 0.06 ^###†^
vWAT (cm^3^)	0.45 ± 0.08	0.56 ± 0.10 *	0.50 ± 0.08	0.22 ± 0.03	0.61 ± 0.07 ****	0.49 ± 0.09 ^###†^
tWAT (cm^3^)	1.41 ± 0.09	1.76 ± 0.12 **	1.58 ± 0.08 ^#^	0.70 ± 0.05	1.86 ± 0.16 ***	1.56 ± 0.19 ^###†^
Tumor (cm^3^)	0.50 ± 0.08	0.42 ± 0.07	0.44 ± 0.06	1.49 ± 0.20	0.96 ± 0.08 *	1.19 ± 0.15 ^#^
Ascites (cm^3^)	0.40 ± 0.47	0.09 ± 0.18 *	0.19 ± 0.19	3.00 ± 0.51	1.75 ± 0.76 *	1.91 ± 0.41 ^#^

*: PC-NS vs. PC-VCP; ^#^: PC-NS vs. PC-SB; ^†^: PC-VCP vs. PC-SB. The number of symbols indicates the size of the *p* value. One symbol (*/^#^/^†^) is *p* less than 0.05, two symbols (**) are *p* less than 0.01, three symbols (***/^###^) are *p* less than 0.001, and four symbols (****) are *p* less than 0.0001. eWAT: epididymis white adipose tissue, sWAT: subcutaneous white adipose tissue, vWAT: visceral white adipose tissue.

**Table 2 nutrients-14-03013-t002:** MRI-measured fat fractions of sWAT, vWAT and eWAT at different time points of treatment.

Fat Fractions (%)	Day 8			Day 16		
	PC-NS	PC-VCP	PC-SB	PC-NS	PC-VCP	PC-SB
sWAT	80.08 ± 0.92	82.43 ± 3.69	81.71 ± 1.30	69.67 ± 3.10	80.48 ± 0.92 **	72.26 ± 4.65 ^†^
vWAT	75.43 ± 3.41	82.22 ± 0.93 **	77.50 ± 2.88 ^†^	74.48 ± 0.61	77.44 ± 1.14 ***	73.79 ± 1.66 ^††^
eWAT	78.91 ± 2.96	84.37 ± 1.12 *	81.39 ± 2.51	69.72 ± 3.51	82.54 ± 2.70 ***	76.54 ± 1.43 ^#††^

*: PC-NS vs. PC-VCP; ^#^: PC-NS vs. PC-SB; ^†^: PC-VCP vs. PC-SB. The number of symbols indicates the size of the *p* value. One symbol (*/^#^/^†^) is *p* less than 0.05, two symbols (**/^††^) are *p* less than 0.01, and three symbols (***) are *p* less than 0.001. CSSI: chemical shift selective imaging, eWAT: epididymis white adipose tissue, FF: fat fraction, sWAT: subcutaneous white adipose tissue, vWAT: visceral white adipose tissue.

**Table 3 nutrients-14-03013-t003:** Weight change in mice with and without the p38 MAPK inhibitor treatment.

	PC-NS	PC-VCP	PC-SB
Net weight (g)	18.88 ± 1.00	20.92 ± 1.05 ***	19.38 ± 1.78 ^†^
Net weight change (g)	−1.61 ± 0.59	0.17 ± 0.99 ***	−0.01 ± 1.09 ^##†††^
Percentage of change (%)	−6.71 ± 1.18	0.11 ± 3.77 ***	−2.02 ± 3.36 ^###†††^

*: PC-NS vs. PC-VCP; ^#^: PC-NS vs. PC-SB; ^†^: PC-VCP vs. PC-SB. The number of symbols indicates the size of the *p* value. One symbol (^†^) is *p* less than 0.05, two symbols (^##^) are *p* less than 0.01, and three symbols (***/^###^/^†††^) are *p* less than 0.001.

## Data Availability

The data that support the findings of this study are available from the corresponding author upon reasonable request.

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
