# Peer review of "Reducing White Adipose Tissue Browning Using p38α MAPK Inhibitors Ameliorates Cancer-Associated Cachexia as Assessed by Magnetic Resonance Imaging"

_nutrients, 2022, doi:10.3390/nu14153013_

Round 1

Reviewer 1 Report

This study indicates that MRI noninvasive quantification of subtype fat distributions combined with p38 MAPK inhibitor  has potential clinical therapeutic application treating patients that present

 cancer-associated cachexia (CAC). The application of MRI and the effect of p38 MAPK inhibitors on the progression of browning is investigated. The manuscript suggests that  imaging and chemicals that inhibit certain molecules  of the p38 pathway are clinically useful in treatment paradigms of patients CAC. The title and abstract are reasonable. The results section is acceptable. The conclusion identifies some of the limitations of the study.

There are several concerns that should be addressed before the study is ready for publication. In the event the authors address the concerns, changes in the manuscript should be highlighted for re-review.

The study examines lipid metabolism and white adipose tissue browning in mice that exhibit pancreatic cancer. The exclusion of female mice in the study should be explained.

The introduction should expand rationale for co-cultures

The rationale for coculturing Panc02 cells and primary white adipocytes should be made clear in the introduction.

Define complete medium used for culture of the various cells.

Has the in vitro cachexia model used in the study been characterized, verified and published?  What is the mechanism underlying the effect of “differentiation induction medium” on cells? Is the medium 50% Panc02 medium plus 50% fresh complete medium?

Cells were cultured in wells of 6-well plates for 6 days. The cell populations were at which phase of growth at the time of data point collection?

Have the orthotopic pancreatic tumor models been characterized? In addition to the supplementary file information on preparation of the pancreatic cancer mouse model used in the study, a reference of the procedure should be included in the main body of the manuscript.

Are the parameters of MRI scans and image analysis standard in the imaging field? References citing the technique to measure WAT should be added.

Define adipogenic cocktail.

Define basic  culture

The statement that the culture supernatant plus Panc02 cells were added to primary white adipocytes should be explained. Were Panc02 cells actually added, as opposed to their supernatant?

Figure 3 title is overstated: The data show the effect of p38 MAPK inhibitor on expression of UPC1, p-38 and lipid droplets.

A summary should be provided for Table 1 and Table 3.

Reviewer 2 Report

Dear authors, 

Overall work here, I believe the authors have done an exemplary job in preparing this manuscript. The level of scientific rigor is apparent, and the attention to detail with regards to every aspect of the replication is appreciated. 

I have a few minor suggestions that the authors might consider, but all of them would moving forward. 

This study is good and needs to be improve for our audience. It contains more lack of literature evidence which is not possible to complete the story here and validated methodology and need to be improve further to build this study strong enough to publish. 

Reviewer 3 Report

This manuscript shows the effect of p38alpha inhibitor in cancer cachexia of pancreas cancer model.

The authors perform the administration of p38alpha inhibitor in vivo with transplanted mice.

Their results show that p38alpha inhibitor improves prognosis in the mouse model.

The reviewer believes that the authors evaluated important issues.

While their results are interesting; however, there are several concerns in the manuscript.

Comments

1. In all bar-plot, the authors should show the individual sample as an overlaid dot.

2. Their results and conclusions confused to the reviewer.

It is unclear p38 inhibition is directly associated with decreasing of cachexia, as claimed by the authors. 

In Figure 4, tumor volume is decreased by inhibitor treatment.

Which does p38 inhibitor in vivo inhibit the differentiation of adipocyte or the cachexia inducing activities in pancreatic cancer cells?

3. Alpha subunit is important for differentiation and it has the potential to be a therapeutic target. It is an interesting, but the mechanism remains unclear.

The authors should discuss the upstream and downstream of p38alpha subunit, form cancer cell to UCP1 expression.

4. In general, cancer cachexia mainly causes poor prognosis by skeletal muscle weakness and anorexia.

These differentiations of adipocyte dramatically affect the prognosis?

This concern makes us difficult to understand the effect of p38alpha inhibitor.

A rational explanation is needed.

5. The manuscript has several mistakes in the character size and font.

Round 2

Reviewer 2 Report

revised well

Reviewer 3 Report

The authors addressed all my concern.